# Improved Water Repellency and Dimensional Stability of Wood via Impregnation with an Epoxidized Linseed Oil and Carnauba Wax Complex Emulsion

**Jinyu Chen, Yujiao Wang, Jinzhen Cao**  **and Wang Wang \***

MOE Key Laboratory of Wooden Material Science and Application, Beijing Forestry University, Beijing 100083, China; chenjinyu@bjfu.edu.cn (J.C.); wyj1996@bjfu.edu.cn (Y.W.); caoj@bjfu.edu.cn (J.C.)

\* Correspondence: wangwang_1987@bjfu.edu.cn

**Abstract:** Natural wood is susceptible to moisture during its practical use, limiting not only service life but also the range of uses. In this study, plant extracts, specifically linseed oil and carnauba wax (both environmentally friendly and water-repellent substances), were examined as a means to mitigate limitations of natural wood. Stable and homogenous epoxidized linseed oil/carnauba wax emulsions with an average particle size less than 230 nm were used as a modifier to enhance the performance of wood. Weight percentage gain, bulking coefficient, micromorphology, chemical structure, moisture adsorption, contact angle, water repellency, and dimensional stability of treated wood were measured. Using a scanning electron microscope (SEM), it was observed that carnauba wax and epoxidized linseed oil acted as a mechanical barrier and could partially cover or block vessels, pits, and wood rays. Furthermore, the hydrophobicity of wood increased significantly after emulsion impregnation. However, there was no obvious chemical reaction between epoxidized linseed oil and the main components of wood in the Fourier transform infrared spectroscopy (FTIR) results. The combined effect of epoxidized linseed oil and carnauba wax was apparent in both decreased water absorption and dimensional deformation. Poplar wood impregnated with an epoxidized linseed oil/carnauba wax complex emulsion shows potential for improving water-related properties. Thus, the complex emulsion modification can improve the inherent shortcoming of poplar wood effectively.

**Keywords:** epoxidized linseed oil; carnauba wax; emulsion; water repellency; dimensional stability

## 1. Introduction

Owing to natural advantages, such as renewability, processability, superior strength-to-weight ratio, and heat insulation, wood has long been used in furniture, decking, cladding, and wooden frames. However, wood is a hygroscopic, anisotropic biopolymer, mainly formed by cellulose, hemicellulose, and lignin. The moisture interaction with wood fiber composites may result in dimensional instability [1–3], which limits its applications under certain environmental conditions.

Water repellent treatments could increase water repellency and dimensional stability of wood, as well as reduce wood checking increased by outdoor weathering. The most common water repellent applied to the wood industry is paraffin wax [4]. The paraffin wax deposited in wood capillaries can reduce water penetration by capillary action, thus limiting dimensional swelling [5]. However, the failure of protection is associated with the nonbond between the cell wall and deposits [6]. A superhydrophobic surface provided by microstructure could resist the contact between water and the wood surface [7]. The waterproof effect relies entirely on the fabrication of the structure on the wood surface, but this microstructure might be damaged when exposed to environmental weathering.

More than that, the hygroscopicity can be minimized by the decreased or covered hydroxy achieved via chemical modification of wood, such as acetylation [8], esterification [9], furfurylation [10], and so on. However, the chemical modifications might be somewhat environmentally unfavorable [11]. It is well-known that there is an increasing need for green and renewable materials, such as waxes and oils, derived from plants [12].

Linseed oil (LO) is a hydrophobic, ecofriendly, non-expensive product derived from dried seeds of flax plant (*Linum usitatissimum* L.) [6]. As a drying oil, LO can block the lumen of the wood and form a stable LO film on pore surfaces after entering the interior of the wood by conventional impregnation techniques [13]. LO could reduce water uptake and increase biological resistance of wood [6,14]. Since no chemical linking is established between wood and oil, the LO impregnation could only reduce the water adsorption rate but not the final moisture content [15]. Furthermore, the oxidative polymerization of LO needs extensive time, resulting in oil to exude from the wood [16]. Linseed oil that has been epoxidized (ELO) may speed up oxidative polymerization [17]. Compared with wood treated with LO, ELO-treated wood has shown significant performance improvements in anti-swelling efficiency (ASE), water repellency, biodegradation resistance, and leaching resistance [18–20]. The main improvement mechanism of ELO-modified wood was correlated to the covalent bonding to the wood cell wall after the epoxy ring opening [19,21]. ELO impregnation seems to be a feasible process for water repellent treatment. However, the hydrophobicity may be restricted by intrinsic properties of polymerized ELO with a contact angle less than 90° [22]. Besides, new hydroxy and carbonyl groups are generated during the polymerization reaction of ELO [19], which is likely to have negative effects on wood surface wettability.

Carnauba wax (CW) is the hardest natural wax, extracted from the leaves of carnauba (*Copernicia prunifera* (Miller) H. E. Moore) [23]. Consisting of esters, straight-chain primary alcohols, hydroxy fatty acids, and other chemical components, it is used commonly for a wide range of applications in the food industry [23], cosmetics [24], and in medical products [25]. CW has been frequently employed in wood modification due to its hydrophobicity and good dispersibility. The deposition of wax particles onto wood surface introduced additional roughness at the micron level [26]. Therefore, the CW-treated wood has been shown to have a water contact angle to be higher than 120°, as well as improved moisture buffering based on noncontinuous coating of CW particles [27]. The wax film is durable, and consecutive thermal treatment leads to melting of the wax particles, even causing no decrease in the hydrophobicity [28]. It can be concluded that wood coated with a CW emulsion may repel water well. Polymerized ELO could bond with wood composites, but shows a low water contact angle. CW could decrease the wettability of wood significantly, but it only works as a mechanical barrier. Thus, incorporating CW into ELO may further improve the hydrophobicity of wood. Both ELO and CW could be easily dispersed in surfactant [29,30], making an ELO/CW complex emulsion a feasible choice.

In China, planted forests have been increasingly utilized due to the logging bans of natural forests. Poplar wood, as one of the most economic and fast-growing species, has been used in packaging, pulping, papermaking, construction production, etc. Unfortunately, undesirable drawbacks of unmodified poplar wood, including poor dimensional stability and durability, restricts its full utilization. In this study, poplar wood (*Populus cathayana* Rehd) was modified using a synthesized ELO/CW complex emulsion to enhance its water-related performance. Following the characterization of chemical components and stabilization of current synthesis formulations, it was impregnated with the emulsion via vacuum pressure. The equilibrium moisture content, water contact angle, water absorption rate, and dimensional stability of the modified wood were subsequently evaluated.

## 2. Materials and Methods

### 2.1. Materials

The samples were cut from poplar wood (*Populus cathayana* Rehd), logged from plantations in Liaoning province, China. The sapwood from the samples was sawed mechanically in the dimensions of $20 \times 20 \times 20$ mm$^3$ (tangential × radical × longitudinal). The air-dried density of detect-free experimental specimens was 0.42 g/cm$^3$ and the average growth ring width was 3.7 mm. There were eight groups in the study of a total of 128 specimens. These specimens were oven-dried at 103 °C to constant weights and dimensions. Commercially available linseed oil (LO), with an initial iodine number of 160.1 g/100 g, and carnauba wax (CW), with a melting point of 80–86 °C, were used. Formic acid (FA), hydrogen peroxide (H$_2$O$_2$), and acetic acid (AA) were used as catalysts for synthesis and the ring-opening of epoxidized linseed oil (ELO). Sodium chloride (NaCl) and sodium bicarbonate (NaHCO$_3$) were used for the purification of ELO. Cyclohexane (CYH), potassium iodide (KI), and sodium thiosulfate solution were used for titration. They were obtained from Beijing Chemical Works Co., Ltd. Alkylphenol ethoxylates (APE-40 and APE-4) purchased from Shandong Usolf Chemical Technology Co., Ltd. were used as the surfactants for emulsion preparation.

### 2.2. Synthesis of Epoxidized Linseed Oil (ELO)

LO (300 g), contained in a three-necked flask, was placed in a constant temperature water bath. A mixture of FA (45 g) and H$_2$O$_2$ (300 g) was added to the flask in drop-wise manner, at approximately 10–15 mL/min, under mechanical stirring at 30 °C. After the FA and H$_2$O$_2$ mixture was consumed, the epoxidation reaction was continued at 52 °C for an additional 5 h. The organic layer was separated from the cooled mixture and then washed repeatedly with a saturated NaCl solution warmed to 45–50 °C, followed by 2% NaHCO$_3$ to remove free acid. The final product was dehydrated by vacuum evaporation and then analyzed for chemical composition and iodine value.

Fourier transform infrared spectroscopy (FTIR) was measured by an FTIR spectrometer (Nicolet 6700 Thermo Scientific, Waltham, MA, America) in wavelengths ranging between 4000 to 600 cm$^{-1}$ with a resolution of 4 cm$^{-1}$ after 32 scans. The spectra of LO and ELO were detected by adding a drop of the oil sample on the infrared probe to investigate the chemical group. The spectra were normalized to the peak at 1460 cm$^{-1}$ (CH$_2$ scissors deformation vibration) [19].

The iodine value, i.e., grams of iodine that can be absorbed by 100 g of sample [31], represented the desaturation of fats and oils. The iodine value of LO and ELO were determined by the Wijs method according to ASTM D5554-15 [32]. Briefly, LO and ELO were dissolved in a mixture of CYH and AA prior to the addition of the Wijs reagent. After placing the solution in a dark environment, KI solution and deionized water were added. Finally, the sodium thiosulfate solution was added to titrate-liberated iodine, using starch as the indicator. Conversion ratio (CR) of the resultant product was analyzed with the iodine number of LO and ELO [29], and the CR was calculated using the following equation:

$$\text{CR } (\%) = \frac{IV_0 - IV}{IV_0} \times 100 \tag{1}$$

where $IV_0$ is iodine number of LO, and $IV$ is the iodine value of ELO.

### 2.3. Preparation and Characterization of Emulsion

All developed formulations were based on an oil-in-water (o/w) emulsion. ELO or CW was mixed with the suitable amount of APE-40 and APE-4 and melted in a beaker. Heated water was added to the mixture under electric agitation, and placed in a high-pressure homogenizer for 10 min. These formulations were then diluted with deionized water to the required concentration and the necessary amount of catalyst introduced. The ELO:AA ratio was 70:30 (w/w). These dilution emulsions were labeled as ELO20, CW2, CW4, CW6, E20/C2, E20/C2, and E20/C6.

The centrifuge stability, particle size, and viscosity of emulsions were tested at ambient temperature. Centrifuge stability was evaluated by using an acceleration centrifuge (LD5-10, Beijing Jingli Freelance Machine Co. Ltd., China) following a previous study [33]. The viscosity of emulsions was assessed using a viscometer (DV-1, Shanghai Fangrui Instrument Co. Ltd., China). The average particle size of diluted emulsions (2–6%) was determined using a laser particle size analyzer (Delsa Nano C, Beckman Coulter, Brea, CA, America).

### 2.4. Wood Impregnation

The specimens were impregnated with dilution emulsions using an identical full-cell impregnation process. After subjecting them to a vacuum of −0.1 MPa for 30 min, the specimens immersed in the dilution emulsions were introduced at a subsequent pressure of 0.5 MPa for 2 h. After being wiped with tissue paper, specimens were dried in ambient room conditions for 4–5 days and then oven-cured at 80 °C for 15 days. Mass and volume of individual specimens were recorded.

The weight percentage gain (WPG) was dependent on the weight dissimilarity before and after impregnation. The bulking coefficient (BC) was dependent on the volume dissimilarity before and after impregnation [34].

The WPG and BC were calculated using the following:

$$\text{WPG } (\%) = \frac{M - M_0}{M_0} \times 100 \tag{2}$$

where $M_0$ is the mass of untreated wood and $M$ is the mass of the treated wood.

$$\text{BC } (\%) = \frac{V - V_0}{V_0} \times 100 \tag{3}$$

where $V_0$ is the volume of untreated wood and $V$ is volume of treated wood.

### 2.5. Micromorphology Characteristic

In order to observe the structure and distribution of modifiers, a scanning electron microscope (JSM-6700F, JEOL Japan) was utilized, with an acceleration voltage of 3–5 kV. Prior to sample preparation, the cubic specimens were cut into small strips and immersed in deionized water. Subsequently, the strips were cut along a cross section and tangential section of the wood using a sliding microtome (REM-710, Yamato Kohki industrial Co., Ltd., Japan). The samples were oven-dried and then secured to a specimen holder with double-sided conductive tape. Fractured surfaces of the samples were coated with gold prior to characterization.

### 2.6. Fourier Transform Infrared Spectroscopy (FTIR) Analysis of Wood

The FTIR spectra were identified with KBr pellet method by using the FTIR spectrometer (Bruker Alpha II, Germany) to investigate the possible chemical change. Prior to FTIR test, the specimens were finely ground to obtain wood powder (200 mesh) and then dried at 80 °C for 2 h. The recorded spectra ranged from 4000 to 400 $cm^{-1}$ with a resolution 4 $cm^{-1}$ after 64 scans. The chosen normalized spectrum had a peak at 1505 $cm^{-1}$ (lignin aromatic ring vibration) after measurement [19].

### 2.7. Equilibrium Moisture Content (EMC)

The moisture adsorption was determined with 5 replicates for each group according to GB/T 1931–2009 [35]. The untreated and treated specimens were conditioned in a climate chamber at 20 °C and 65% relative humidity (RH) for 4 weeks to the constant weight. The EMC was calculated according to:

$$\text{EMC } (\%) = \frac{M_e - M}{M_0} \times 100 \tag{4}$$

where $M_e$ is the weight of specimens after being conditioned in a climate chamber.

## 2.8. Water Contact Angle (WCA) Characterization

Wettability of the wood transversal surface was determined via contact angle meter (SL200KS, USA KINO Industry Co. Ltd., USA) using a 4 µL droplet of deionized water. The sanded wood specimens (1500 mesh sandpaper sanding for 5 min before impregnation) were preconditioned at 20 °C and 65% RH for two weeks. The time-dependent WCA was recorded once per second for a total of 80 s. The result was an average value of 5 replicates of samples.

## 2.9. Water Uptake and Swelling Test

For determination of water absorption rate (WA) and anti-swelling efficiency (ASE) of wood, a total of 40 samples (5 replicates per group) were prepared. Samples were immersed in deionized water at room temperature. The wet weights and dimensions were measured after 1, 6, 12, 24, 48, 96, and 192 h immersion periods [3]. The WA and ASE were calculated according to:

$$\text{WA } (\%) = \frac{M_i - M}{M_0} \times 100 \tag{5}$$

$$\text{S } (\%) = \frac{V_i - V}{V} \times 100 \tag{6}$$

$$\text{ASE } (\%) = \frac{S_C - S_T}{S_C} \times 100 \tag{7}$$

where $M_i$ is the weight and $V_i$ is volume after soaking. $S_C$ and $S_T$ represent the volumetric swelling coefficients of control wood specimens and treated wood, respectively.

## 3. Results and Discussion

### 3.1. Characterization of ELO and Emulsions

The iodine value reflects the degree of unsaturation of vegetable oils, and the larger iodine value reflects the more carbon-carbon double bonds per vegetable oil triglyceride [36]. Therefore, the iodine value can be used to evaluate the conversion efficiency of the epoxidation reaction.

In situ epoxidation of linseed oil in the presence of catalysts is a two-step functional group conversion process. Peroxyacid formation is followed by an epoxy group formation, which is produced by the reaction of an unsaturated double bond with peroxyacid [37,38]. The FTIR spectra of LO and ELO are shown in Figure 1. The band at 3011 cm$^{-1}$, which represented a carbon-carbon double bond [39,40], disappeared after epoxidation. A new band at 820 cm$^{-1}$, attributed to epoxy groups [40,41], was found in ELO, indicating that the unsaturated bond had been converted. In addition, the iodine value decreased from 160.1 g/100 g of LO to 8.2 g/100 g of ELO. The calculated conversion ratio was 94.8%, suggesting the epoxidation reaction in this study reached a high conversion efficiency.

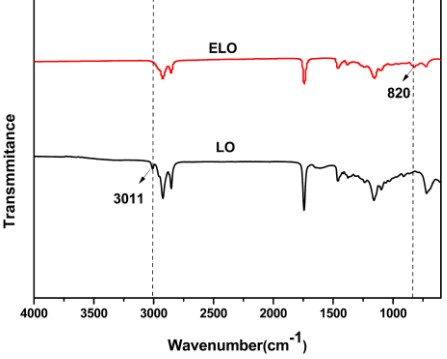

**Figure 1.** Fourier transform infrared spectroscopy (FTIR) spectrum of linseed oil (LO) and epoxidized linseed oil (ELO).

Figure 2 illustrates the stable and uniform appearance of the emulsions. There was no phase separation nor uneven viscosity, and all were uniformly white or light-yellow in color. The most common instability within emulsions occurred during the dispersed-phase separation, due to the density dissimilarity between phases. Centrifugation may speed up the collisions between globule the emulsion particles [42]. Thus, to better estimate the stability of emulsion during storage, the particle size and viscosity were detected before and after centrifugation. Average particle diameters were no more than 230 nm (Table 1), and all of the particle diameters were smaller than those of the vessel, wood fibers, and wood rays, as reported by Carlquist [43]. Thus, the prepared emulsions could penetrate easily into the interior of the wood. A slight increase in particle diameters was observed after centrifugation (< 250 nm), and the maximum particle size change rate was 2.6% for the E20/C4 emulsion. The emulsions retained a consistent constant outward appearance and the viscosity had no noticeable change after centrifugation.

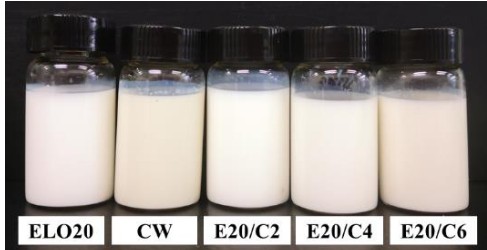

**Figure 2.** The appearance of different emulsions.

**Table 1.** The ELO or carnauba wax (CW) content, particle size, and viscosity before and after centrifugation of emulsions.

| Groups | ELO Content (%) | CW Content (%) | Particle Size | | Viscosity | |
|---|---|---|---|---|---|---|
| | | | (Before) (nm) | (After) (nm) | (Before) (mPa·s) | (After) (mPa·s) |
| ELO20 | 20 | - | 195.0 | 194.2 | 45 | 52 |
| CW2 | - | 2 | 219.8 | 222.5 | 22 | 22 |
| CW4 | - | 4 | 219.8 | 221.9 | 22 | 23 |
| CW6 | - | 6 | 220.3 | 225.7 | 23 | 22 |
| E20/C2 | 20 | 2 | 195.3 | 197.5 | 26 | 27 |
| E20/C4 | 20 | 4 | 188.8 | 205.1 | 28 | 28 |
| E20/C6 | 20 | 6 | 229.9 | 237.9 | 30 | 30 |

### 3.2. Weight Percent Gain (WPG) and Bulking Coefficient (BC)

The WPG of ELO emulsion-treated specimens were 30.95%. Table 2 illustrates that the WPG of wood treated with CW emulsions was proportional to the content of CW. However, there was no significant difference between the WPG of the samples treated with complex emulsions that had differing CW content. Compared with the sum of WPG obtained from the corresponding ELO and CW emulsion-treated groups, the ELO/CW emulsion-treated groups showed a lower WPG (except for the E20/C2 group), suggesting that WPG was limited by the penetration threshold of wood.

BC of ELO emulsion-treated specimen was 2.45%, as presented in Table 2. The effect of the CW emulsion-based treatment on BC of wood was notably inferior to that of the ELO emulsion-based treatment. However, there was no distinct difference in BC values among samples treated by ELO/CW complex emulsions with different solid contents. We hypothesize that modifiers are primarily filling the wood lumen, but not further bulking the cell wall in the treated specimens [44].

**Table 2.** The weight percent gain (WPG), bulking coefficient (BC), and equilibrium moisture content (EMC) of untreated and treated wood.

| Groups | WPG (%) | BC (%) | EMC (%) |
|---|---|---|---|
| control | −0.52 ±0.03 | 0.56 ± 0.23 | 8.46 ± 0.55 |
| ELO20 | 30.95 ± 1.31 | 2.45 ± 0.38 | 8.11 ± 0.16 |
| CW2 | 3.67 ± 0.55 | 0.57 ± 0.35 | 8.28 ± 0.07 |
| CW4 | 6.98 ± 0.53 | 0.89 ± 0.49 | 8.28 ± 0.14 |
| CW6 | 9.93 ± 1.27 | 1.17 ± 0.49 | 8.24 ± 0.06 |
| E20/C2 | 36.06 ± 1.82 | 2.71 ± 0.25 | 7.89 ± 0.66 |
| E20/C4 | 34.65 ± 1.91 | 2.11 ± 0.61 | 7.76 ± 0.05 |
| E20/C6 | 35.80 ± 1.14 | 2.48 ± 0.25 | 7.59 ± 0.27 |

### 3.3. Morphological Observation of Wood

Comparing within the same treated group, there was no significant difference among wood treated with varying emulsion concentrations. Therefore, only different types of SEM results are shown in Figure 3. The cell lumens, pits, and ray cells were empty and visible in the untreated wood (Figure 3a,e). The results demonstrate that cell lumens are partially filled after impregnation with the different emulsions (Figure 3b–d), and that the pits and ray cells are covered or blocked by CW and ELO (Figure 3f–h).

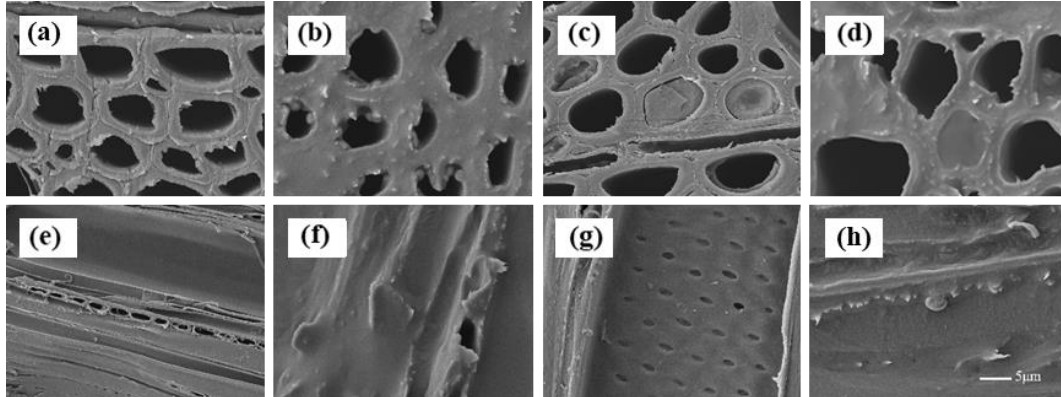

**Figure 3.** SEM observations on the transverse and tangential sections of untreated wood (**a** and **b**), ELO20-treated wood (**c** and **d**), CW4-treated wood (**e** and **f**), and E20/C4-treated wood (**g** and **h**).

Moreover, there were a few wax particles attached to cell walls (Figure 3g) in the CW emulsion-treated wood, as viewed in tangential sections. Wax particles (Figure 3h) might be partially covered by oil layers in the ELO/CW emulsion-treated wood. Unlike the lumen of CW emulsion-treated wood, the lumens in the ELO-treated wood were blocked by solidified modifiers, as shown in the transverse section (Figure 3b,d). Perhaps this is because emulsified CW has a lower viscosity than ELO, thus facilitating improved diffusion into the wood. This could also explain the more uniformly layered seal on the lumen surface (Figure 3d), despite the higher WPG, as demonstrated in Table 2.

These results confirm that various emulsions can penetrate the wood through vessel, pits, and rays. Continuous and discontinuous layers block the transport path of free water, especially within the longitudinal pathways of emulsion-treated wood. The small amount of granular structure may reduce the possible contact between free water and wood, thus enhancing the hydrophobicity of the wood.

### 3.4. Chemical Structure Analysis Using FTIR Spectroscopy

Variations in chemical functional groups of treated wood were detected by FTIR. The structures of both CW and ELO contain long alkyl chains, therefore asymmetric and symmetric stretching peaks representing C-H ($CH_3$ and $CH_2$) at 2921 and 2852 $cm^{-1}$ appeared in the treated specimens (Figure 4).

AA was added to the ELO emulsion and ELO/CW emulsion formulations, and thus the spectral results were more complicated than those of the CW emulsion-treated wood. The bands at 1375 and 1740 cm$^{-1}$ correspond with a methoxy group and C=O, respectively [19,45]. The enhancement of band intensities at 1740 cm$^{-1}$ is attributed to an intrinsic triglyceride structure and the grafting of acetic acid onto epoxidized linseed oil [19]. The reaction mechanism between wood and ELO has been discussed previously [21,29,46]. ELO occupied the hydroxyl sites of wood and produced new ether functions 1300–1193 cm$^{-1}$ [19]. Contrary to expectations, there was no increased intensity in bands at 1200–1120 cm$^{-1}$ attributed to the ether bond peak, which suggests no chemical reaction between ELO and wood. This difference may be attributed to the specific formulation system and to oil loading.

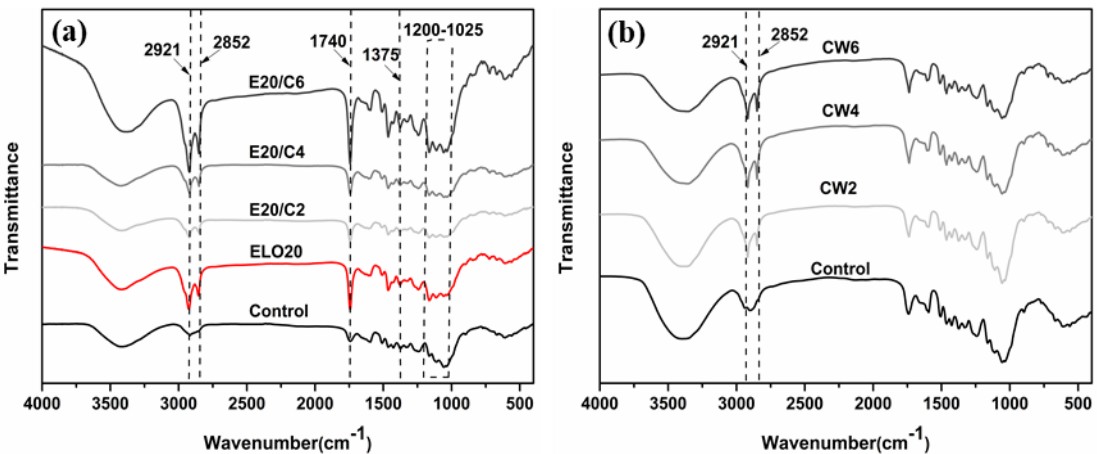

**Figure 4.** The FTIR spectrum of untreated wood and carnauba wax emulsion-treated wood (**a**), and ELO emulsion-treated wood and complex-emulsion treated wood (**b**).

### 3.5. Equilibrium Moisture Content (EMC)

Table 2 demonstrates the EMC after conditioning in a chamber at 20 °C and 65% RH. The EMC of untreated wood was 8.46%. Water vapor adsorption in the cell wall was decreased by CW and ELO. Compared with untreated wood, the EMC of ELO emulsion-treated wood, CW emulsion-treated wood, and ELO/CW emulsion-treated wood averaged 8.11%, 8.26%, and 7.75%, respectively. Although the treatment did not change the chemical structure of wood, the ELO/CW emulsion markedly decreased the vapor adsorption of treated wood, and therefore enhanced dimensional stability.

### 3.6. Hydrophobicity of Wood

The dynamic water contact angle (WCA) of treated and untreated wood are illustrated in Figure 5. The initial WCA of untreated wood was 127° and it consistently fell to 0° within 80 s. However, the WCA for CW emulsion-treated wood was more stable. It ranged between 135-140°, indicating that a hydrophobic water barrier on the wood surface was initiated by CW. The contact angle is affected by dominant factors that include surface roughness and surface free energy [47]. The natural roughness of wood, combined with the low surface energy of the CW granular structure (Figure 3g), made the CW emulsion-treated wood more hydrophobic [27].

As described in previous work, the WCA of polymerized ELO decreases slowly, from 50° to 38° within 30 s [48]. However, the ELO emulsion-treated wood showed a rapid decrease in WCA, from 80° to 0° within 23 s. This rapid decrease in WCA might be attributed to the enhancement of the hydrophilic carbonyl group (Figure 4a) after the ELO film polymerized on the wood surface. Although the complex emulsion-treated wood did not perform as well as the CW emulsion-treated wood, the initial WCAs are all above 120°, demonstrating a slow decrease, but were maintained over 80° after 80 s. The performance may be strongly influenced by decreased surface energy. Additionally, the intensity of the carbonyl group observed from ELO/CW emulsion-treated wood was lower than

that of the ELO emulsion-treated wood (Figure 4a), which might contribute to the less hydrophilic wood surface. When the CW content was 4%, the WAC remained above 120° after 80 s.

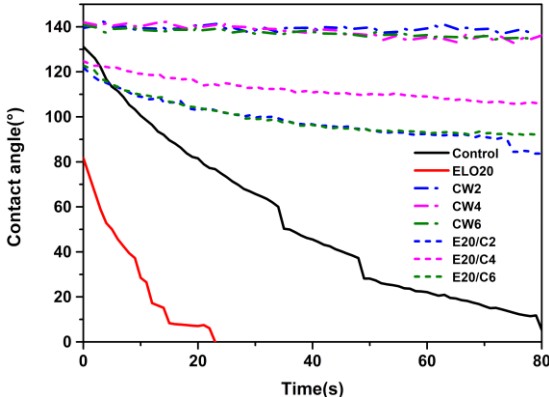

**Figure 5.** Water contact angle (WCA) of treated and untreated wood on transverse sections as a function of time.

### 3.7. Liquid Water Uptake and Dimensional Stability

Both vegetable oil emulsion and wax emulsion have been proven to be efficient water repellents [49,50]. Three kinds of emulsions in this study showed differential effects of inhibiting water absorption (Figure 6a). The WAs of untreated wood were approximately 27% and 132% after soaking for 1 and 192 h, respectively. The WA of CW emulsion-treated wood was approximately 53% lower than untreated wood after soaking for 1 h, but the WA increased to 129% after 192 h. This result indicates that CW emulsion only has short-term effectiveness. CW forms a hydrophobic film-barrier on wood surface, which could separate the contact between the wood surface and water to slow water movement [33,51]. However, there are no chemical reactions occurring between CW and the hydroxyl groups of wood [52]. Thus, film-barriers may be lost during long periods of immersion, allowing water molecules to swell the wood, opening up the inaccessible region of wood cell walls to water absorption [53]. ELO emulsion-treated wood seems to have better water resistance than CW emulsion-treated wood, although it has a poor WCA. The ELO/CW emulsion-treated wood exhibited the lowest value of WA among all groups. Waterproofing of the wood is related to the tendency of the wood surface to be resistant to interactions with water [52]. Although ELO has no reaction with hydroxyl groups of wood, the ELO could open its epoxy ring and react with AA (Figure 4b). The reaction would produce inter- and intramolecular crosslinking [54], which results in a denser, more durable film than with the CW membrane.

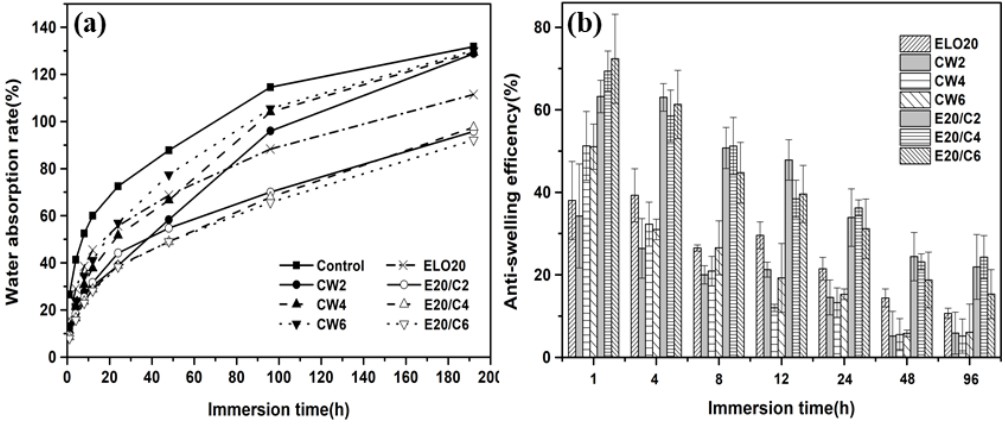

**Figure 6.** Results of water absorption rate (WA) (**a**) and anti-swelling efficiency (ASE) (**b**) of untreated wood, CW emulsion-treated wood, ELO emulsion-treated wood, and ELO/CW emulsion-treated wood.

This study explores the synergic effect of mixing hydrophobic CW and polymerized ELO. ELO can form macromolecular material via self-polymerization and form a complete film on a wooden surface. However, due to the relatively high surface energy of the ELO film, wood surfaces are more prone to forming attachments with water molecules, which exposes the ELO film to long-term contact with moisture and oxygen. This action results in accelerated weakness of the mechanical bonding between ELO and wood components, and may cause aging or a cracking of the ELO polymer film. When CW is added, the surface energy of the film is significantly lower. The wood surface is no longer vulnerable to moisture attachment, which reduces the possibility of microcracks in the ELO polymer film. Hence, water molecules cannot penetrate into wood spontaneously via capillary action, thus restricting the rate of water absorption [49].

ASE results are demonstrated in Figure 6b. The ASE of CW emulsion-treated wood after 1 h is 50%, whereas that of ELO emulsion-treated wood is 38%. However, the ASE value of ELO emulsion-treated wood is larger than the CW emulsion-treated wood with increased immersion time. Obviously, there is a reduction in ASE in all groups with immersion time (Figure 6b), but the ELO/CW emulsion-treated wood shows the best dimensional stability, with a maximum ASE of 15.3% after 192 h. There are two factors that may relate to this improved dimensional stability.

One is that the ELO/CW treatment creates a mechanical hindrance to free water absorption. This explanation is supported by the water absorption test, which demonstrates lower average WA of ELO/CW emulsion-treated wood (95%) after 192 h. Decreased water absorption could reduce dimensional variation within a short time period [21]. The water-repellent acts to partially block water passage, thereby reducing water's capacity to penetrate the wood structure.

Previous reports have suggested that emulsion particles (>100 nm) were too large to penetrate cell walls [40], whereas our data suggest a different result. The cell wall may partially fill with ELO, providing the other explanation for moderate ASE. Similar results have been reported by Panov et al. [21], in which the ELO penetrated the wood structure and was present in the wood cell walls.

## 4. Conclusions

Impregnation of newly prepared ELO/CW emulsions effectively improved the water-related performance of wood. The emulsions had a small particle size (<230 nm) and were stable before and after centrifugation. After impregnation, the main water passages, such as pits and ray cells in wood, were covered or blocked by both CW and ELO, resulting in enhanced wood protection. Adding a small quantity of CW did not contribute to weight percentage gain and not affect the bulking coefficient of treated wood, but decreased the surface energy in complex emulsion systems. The WCA of ELO/CW emulsion-treated wood was as high as 120°, dropping to 80° within 80 s. Although the FTIR analysis showed no evidence confirming the chemical reaction between ELO and wood cell walls, ELO could polymerize itself in the presence of catalyst, forming a dense film on the wood surface. Combining ELO and CW could enhance the water repellency of treated wood, and thus less moisture was absorbed by the treated wood, resulting in an increased ASE. The joint effect was demonstrated in complex emulsion-treated wood with a WA of 92.2% and ASE of 15.3% after 192 h. In conclusion, the ELO/CW emulsion treatment provides sustainable modified wood with excellent water repellency and dimensional stability. This method can provide possibilities to extend the service life of poplar wood.

**Author Contributions:** W.W. and J.C. (Jinyu Chen) conceived and designed the experiments; J.C. (Jinyu Chen) and Y.W. performed the experiments and analyzed the data; J.C. (Jinyu Chen) wrote the original draft; J.C. (Jinzhen Cao) and W.W. reviewed and revised the manuscript. All authors have read and agreed to the published version of the manuscript.

**Funding:** This research was funded by the National Natural Science Foundation of China (No. 31600452) and the Fundamental Research Funds for the Central Universities in China (2019JQ3013).

**Conflicts of Interest:** The authors declare no conflict of interest.

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
