# Peer review of "Improved Water Repellency and Dimensional Stability of Wood via Impregnation with an Epoxidized Linseed Oil and Carnauba Wax Complex Emulsion"

_forests, doi:10.3390/f11030271_

Round 1
Reviewer 1 Report
In the specific scientific field and research area, many researches have been implemented, but they were not included and discussed in the introduction section. Therefore, a descriptive analysis should be included. Otherwise, the innovation of the work can not be proven or highlighted.
In Materials and methods section, there are not information provided on the specimens construction, such as the dimensions of the specimens, the way that they were cut, if sanding process was applied in the specimens before treatment, the average number of annual rings, the exact origin of the specimens material, the exact number of the impregnated specimens, while there are no adequate information provided on the impregnation process itself. All these information are very significant and crucial not only for the design and implementation of the research, but also they are necessary to be provided for the proper evaluation and review process of an experimental work, as well as for the reliability proof of the results.
Author Response
Dear reviewer:
Thank you for time and efforts spent in reviewing the manuscript entitled “Improved water repellency and dimensional stability of wood via impregnation with an epoxidized linseed oil and carnauba wax complex emulsion” (Manuscript ID: forests-726652).
Those comments and suggestions are valuable and very helpful for improving our paper, as well as the important guiding significance to our researches. We have studied comments carefully and have made correction. Revised portion are highlighted in yellow background in the paper. The main corrections in the paper and the responds to the reviewer’s comments are as flowing:
Responses to the comments:
Comments:
In the specific scientific field and research area, many researches have been implemented, but they were not included and discussed in the introduction section. Therefore, a descriptive analysis should be included. Otherwise, the innovation of the work can not be proven or highlighted.
Response:We are grateful for the suggestion. To make the statement more convincing and complete, we have added and discussed some researches on the scientific field we study. And we have enriched the research content with more descriptive analysis in Introduction section.
(Lines 40-49 of page 1-2, Lines 54-57 of page 2, Lines 71-78 of page 2)
In Materials and methods section, there are not information provided on the specimens construction, such as the dimensions of the specimens, the way that they were cut, if sanding process was applied in the specimens before treatment, the average number of annual rings, the exact origin of the specimens material, the exact number of the impregnated specimens, while there are no adequate information provided on the impregnation process itself. All these information are very significant and crucial not only for the design and implementation of the research, but also they are necessary to be provided for the proper evaluation and review process of an experimental work, as well as for the reliability proof of the results.
Response: We are very sorry for our negligence of specific information of the specimens. We have added the required information as follows: “The samples were cut from poplar wood (Populus cathayana Rehd), logged from Liaoning province, China. The sapwood from the samples was sawed mechanically in the dimensions of 20×20 ×20 mm3 (tangential×radical×longitudinal). The air-dried density of detect-free experimental specimens was 0.42 g/cm3 and the average growth ring width was 3.7 mm. There were eight groups in the study of totally 128 specimens.”
(Lines 94-98 of page 3)
The sanded wood that I mentioned in the Water contact angle (WCA) characterization was sanded sandpaper to sand the wood surface actually to avoid the inherent roughness of wood to influence the results water contact angle. I have added the information “1500 mesh sandpaper sanding for 5 minutes”
(Lines 182 of page 4)
We have supplemented more information impregnation process.
“The specimens were impregnated with emulsions using an identical full cell method impregnation process. After subjecting to a vacuum of -0.1 MPa for 30 minutes, the specimens immersed in the dilution emulsions introduced at a subsequent pressure of 0.5 MPa for 2 h”
(Lines 145-147 of page 4)
We tried our best to improve the manuscript and made some changes in the manuscript. And we did some other minor corrections as well. These changes will not influence the framework of the paper. And we hope that the correction will meet with approval. Special thanks to you for your comments and suggestions again.
Yours sincerely,
Jinyu Chen, Yujiao Wang, Jinzhen Cao and Wang Wang

Reviewer 2 Report
The manuscript deals with an interesting topic and focuses on the improvement of the wood properties through its modification/impregnation with linseed oil and carnauba wax. Generally, the manuscript is quite well written, there are not problems with English language or scientific writing, but I would like to highlight and comment on some points where the author's work should be improved, making some changes/additions in order to enhance its quality.
In the Introduction chapter, there are not enough references, the literature and the state of the art is not well described or analysed. There are some previous researches that have used linseed oil or carnauba wax individually as means of wood modification and their results were not taken into consideration.
In materials chapter, the authors did not refer to the number of trees that were cut and included in the experimental work as raw material. Did the species come from plantations or forest area? Which was the density of the wood specimens? Also, it is not explained why they have chosen to test the specific wood species (Populus cathayana Rehd) and which is the significance of this species modification. Therefore, the significance of this work should be more highlighted in the abstract, introduction and conclusions as well. The conclusions chapter is well prepared highlighting the main results and draws a conclusion.
You should refer, probably in materials and methods chapter, the reason why you chose Sodium bicarbonate for the neutralization process, cyclohexane, potassium iodide, and sodium thiosulfate solution for titration, respectively, and provide some references on these. Please, provide reference or standard used for the Weight Percentage Gain, Bulking Coefficient etc.
Author Response
Dear reviewer:
Thank you for time and efforts spent in reviewing the manuscript entitled “Improved water repellency and dimensional stability of wood via impregnation with an epoxidized linseed oil and carnauba wax complex emulsion” (Manuscript ID: forests-726652).
Those comments and suggestions are valuable and very helpful for improving our paper, as well as the important guiding significance to our researches. We have studied comments carefully and have made correction. The main corrections in the paper and the responds to the reviewer’s comments are as flowing:
Responses to the comments:
Comments:
The manuscript deals with an interesting topic and focuses on the improvement of the wood properties through its modification/impregnation with linseed oil and carnauba wax. Generally, the manuscript is quite well written, there are not problems with English language or scientific writing, but I would like to highlight and comment on some points where the author's work should be improved, making some changes/additions in order to enhance its quality.
Response:Thank you very much for your positive comments on the manuscript. We have revised the manuscript carefully according to your suggestions, and all therevisions are highlighted with yellow background in the revised manuscript.
In the Introduction chapter, there are not enough references, the literature and the state of the art is not well described or analysed. There are some previous researches that have used linseed oil or carnauba wax individually as means of wood modification and their results were not taken into consideration.
Response:Thank you for your suggestion. To make the statement more convincing and complete, we have enriched and discussed some researches on the linseed oil and carnauba wax study on wood modification in Introduction chapter.
(Lines 54-57 of page 2, Lines 71-78 of page 2)
In materials chapter, the authors did not refer to the number of trees that were cut and included in the experimental work as raw material. Did the species come from plantations or forest area? Which was the density of the wood specimens? Also, it is not explained why they have chosen to test the specific wood species (Populus cathayana Rehd) and which is the significance of this species modification. Therefore, the significance of this work should be more highlighted in the abstract, introduction and conclusions as well. The conclusions chapter is well prepared highlighting the main results and draws a conclusion.
Response:Regarding the suggestion about the material, We have added the information about specimens as follows: “The samples were cut from poplar wood (Populus cathayana Rehd), logged from plantations in Liaoning province, China. The sapwood from the samples was sawed mechanically in the dimensions of 20×20 ×20 mm3 (tangential×radical×longitudinal). The air-dried density of detect-free experimental specimens was 0.42 g/cm3 and the average growth ring width was 3.7 mm. There were eight groups in the study of totally 128 specimens.” (Lines 94-98 of page 3)
And the reason why we choose poplar wood (Populus cathayana Rehd) has been described in last paragraph of Introduction chapter. (Lines 81-85 of page 2)
Also, we have highlighted the significance of this species modification with sentence “This treatment can be improved the inherent shortcoming of poplar wood effectively, thus potentially extending the economic value of its products.” in abstract (Lines 27-28 of page 1) and sentence “This method can provide possibilities for increasing the value of poplar wood in application due to the improved dimensional stability.” in conclusion. (Lines 365-366 of page 10)
You should refer, probably in materials and methods chapter, the reason why you chose Sodium bicarbonate for the neutralization process, cyclohexane, potassium iodide, and sodium thiosulfate solution for titration, respectively, and provide some references on these. Please, provide reference or standard used for the Weight Percentage Gain, Bulking Coefficient etc.
Response: Thank you to underlining this deficiency. I chose cyclohexane, potassium iodide, and sodium thiosulfate solution for titration according to the ASTM D5554-15 (mentioned in line 122 of page 3). And sodium bicarbonate is used to remove residual acid in the produce process of epoxidized linseed oil. And I have supplied reference or standard for the about the material and Weight Percentage Gain, Bulking Coefficient etc.
(Line 126 for conversion ratio, line 152 for weight percentage gain and bulking coefficient, line 174 for equilibrium moisture content, line 190 for water absorption rate and anti-swelling efficiency )
We tried our best to improve the manuscript and made some changes in the manuscript. And we did some other minor corrections as well. These changes will not influence the framework of the paper. And we hope that the correction will meet with approval. Special thanks to you for your comments and suggestions again.
Yours sincerely,
Jinyu Chen, Yujiao Wang, Jinzhen Cao and Wang Wang
